

# Towards a blockchain-based certificate authentication system in Vietnam

Binh Minh Nguyen, Thanh-Chung Dao and Ba-Lam Do

School of Information and Communication Technology, Hanoi University of Science and Technology, Hanoi, Vietnam

## ABSTRACT

Anti-forgery information, transaction verification, and smart contract are functionalities of blockchain technology that can change the traditional business processes of IT applications. These functionalities increase the data transparency, and trust of users in the new application models, thus resolving many different social problems today. In this work, we take all the advantages of this technology to build a blockchain-based authentication system (called the Vietnamese Educational Certification blockchain, which stands for VECefblock) to deal with the delimitation of fake certificate issues in Vietnam. In this direction, firstly, we categorize and analyze blockchain research and application trends to make out our contributions in this domain. Our motivating factor is to curb fake certificates in Vietnam by applying the suitability of blockchain technology to the problem domain. This study proposed some blockchain-based application development principles in order to build a step by step VECefblock with the following procedures: designing overall architecture along with business processes, data mapping structure and implementing the decentralized application that can meet the specific Vietnamese requirements. To test system functionalities, we used Hyperledger Fabric as a blockchain platform that is deployed on the Amazon EC2 cloud. Through performance evaluations, we proved the operability of VECefblock in the practical deployment environment. This experiment also shows the feasibility of our proposal, thus promoting the application of blockchain technology to deal with social problems in general as well as certificate management in Vietnam.

# INTRODUCTION

The first practical application using blockchain technology is Bitcoin, which is one of the highest valuable crypto-currencies (*Narayanan et al., 2016*). Bitcoin's value has been grown thousands of times since the appearance of the coin, and this is the main reason to explain the popularity of blockchain today. Besides cryptocurrency, blockchain also has emerged as a useful technology, which can be applied to many current social problems caused by the lack of trust among different parties.

In IT applications, a question is casually raised: how to ensure trust with historical data in the system? Under this viewpoint, keeping data integrity is one of the key characteristics that help deal with trust problems nowadays. For centralized applications, data integrity should be guaranteed using secured databases or a trustworthy third party. Meanwhile, for

Corresponding author
Binh Minh Nguyen,
minhnb@soict.hust.edu.vn

decentralized applications, ensuring this feature is a significant challenge, not only because these applications typically contain a limited number of actors should not control many actors but also the data confirmation. Therefore, dealing with data integrity problem has a great potential to be applied to the decentralized applications, which are currently the majority with the development of the Internet.

In Vietnam, using and trading fake certificates is an urgent problem that must be resolved in order to ensure the reliability of the national education system. According to *Tư (2017)*, in 2004, the Vietnamese ministry of education and training detected more than ten thousand fake certificates with other related documents being sold publicly via the Internet, social media and sophisticated trading platforms (*Labour Journal, V, 2018*). Although the Penal Code criminates the use of fake diplomas promulgated in 2015, there are still people using fake documents and seals when applying for jobs. Moreover, other than subjective reasons which spring from a small citizen's group in Vietnamese society, there are also other objective reasons caused by the complexity and overlap in the education management system and the lack of applying modern technologies. Different ministries or national agencies can manage educational institutions in Vietnam at the same time. Besides, with an approximate population of one hundred million, Vietnam thus has a vast number of educational organizations, including schools, colleges, universities that belong to education systems and training centers, which provide short courses for learners. There is a shared database that enables all schools of the basic educational system (primary, secondary, and high schools) to send education records periodically to the ministry. Nevertheless, that database does not integrate with data from universities and other educational organizations (e.g., training centers). Besides, because many actors participate simultaneously in this educational management system, the data created by teachers and schools could be changed by an unwanted third party. Hence, in general, the reliability of certificates and diplomas in Vietnam is very low, so having an educational certification authentication system in Vietnam is an urgent requirement now.

In this study, we focus on building a prototype of an authentication system using blockchain for certificates in Vietnam. We classified and analyzed the blockchain trends to highlight our contributions in the application development area of this technology. We also pointed out the fake certificate problem in Vietnam and the capability of applying blockchain to resolve it based on its functionalities: anti-forgery information, transactional verification, and smart contract. After that, we studied and proposed some development principles for decentralized applications. We presented steps by steps the development of VECefblock: a system for authenticating certificates that ensures the trust of data and provides advantages such as preventing counterfeit diplomas/certificates, supporting integrity feature for stored data. The underlying foundation of VECefblock is blockchain technology, together with a data mapping structure to transfer data from different educational institutions to a unified representation of transactions in the blockchain network. We used Hyperledger Fabric as a blockchain platform, which is configured and deployed on Amazon EC2 service. The achieved performance test results showed the operability of our blockchain-based system in the practical deployment environment. This achieved outcome is one of the important proofs that can be used to promote the

blockchain technology application in resolving social issues in general and certificate management in Vietnam. Specifically, our contributions of this work include:

1. Analyzing the blockchain situations in the aspects of technologies, classifications, functionalities, research trends, and applications;
2. Figuring out the problem of fake certificates in Vietnam and proposing a potential solution using blockchain technology;
3. Proposing principles to develop decentralized applications with a concrete example (VECefblock), in which the system business processes, and data mapping structure are important designs for the development;
4. Designing an educational system for certificate authentication using blockchain technology named VECefblock that suits the situation of Vietnam;
5. Deploying a blockchain platform (Hyperledger Fabric) on public cloud infrastructure (Amazon EC2) together with testing and evaluating the blockchain platform as well as VECefblock's performance and operations;
6. Promoting the use of blockchain technology to resolve social problems in Vietnam.

The rest of this paper is organized as follows. In the next section, a taxonomy of blockchain technologies, applications, and research trends is presented. Next, we introduce the Vietnamese educational system with a fake degree problem. After that, we have a section to introduce system architecture, business processes, and the data mapping structure. We also describe the experimental methodology and achieved test outcomes in the next section. This paper concludes with an outlook on future routes in the last section.

# RELATED WORK

## Blockchain taxonomy

Many studies have analyzed blockchain technologies (*Yli-Huumo et al., 2016*; *Merkaš, Perkov & Bonin, 2020*). In this paper, we build a blockchain taxonomy in order to describe the aspects of blockchain classifications, functionalities, research trends, characteristics, and applications. The main goal of those analyses is to locate our contributions in the overall picture of blockchain at present. Figure 1 illustrates the blockchain taxonomy. In terms of platform purposes, a blockchain can be used for crypto-currency payment Ethereum (*Wood, 2018*) or running distributed application (DApp) (*EOS.IO, S, 2018*). Meantime, a blockchain can be classified by different network types, which is the kind of applications running on it and the chain openness. A network type here refers to the permission of users to read/write the ledger. If anyone can join a blockchain network, it is categorized as a permissionless blockchain. For example, Bitcoin (*Nakamoto, 2008*) and Ethereum (*Wood, 2018*). By contrast, a permissioned blockchain is managed by an organization or a company, who chooses who can read/write a transaction to the blockchain—for example, Hyperledger Fabric (*Androulaki et al., 2018*) and Quorum (*JP Morgan Chase, 2016*). A consortium blockchain stands at the middle point and is defined as a partly private network, which is controlled by a group of individuals or organizations. A consortium network still provides some degree of central control (*Zheng et al., 2017*). In terms of platform purposes, a blockchain can be used for crypto-currency payment (*Wood, 2018*) or running distributed applications (DApp) (*EOS.IO, S, 2018*).

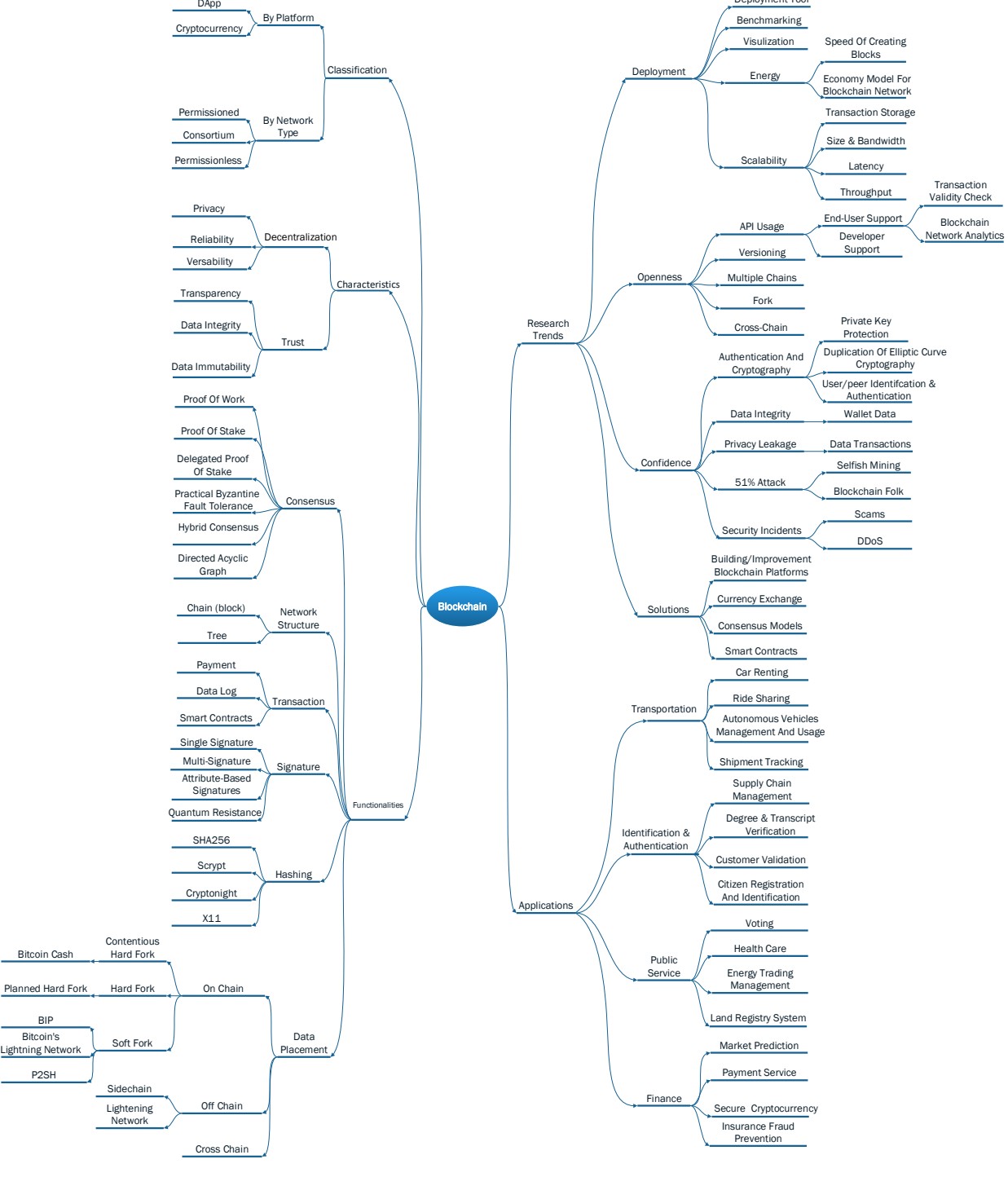

**Figure 1**   **Blockchain taxonomy.**

Because blockchain systems operate on peer-to-peer networks, the main characteristic of blockchain technology is natural decentralization. Thus, transaction data is propagated and stored on all peers/nodes participating in the network system. The combination of decentralization with cryptography algorithms in blockchain brings privacy, reliability, and versatility for its users (*Seebacher & Schüritz, 2017*). In which, privacy is shown through the ability of user anonymity, reliability could be gained because data is stored redundantly on all nodes, and versatility establishes via the capability of developing many applications belonging to different domains based on the technology. On the other side, trust is one of the valuable blockchain characteristics and shown through transparency, data integrity, and immutability feature (*Xu et al., 2017*). In this way, blockchain users have traceability using data recorded in blocks, which is published and shared on all peers. Meanwhile, once data is packed into blocks, it cannot be changed by third-parties. Hence, the integrity and immutability of data can be obtained easily in blockchain systems.

There are components including consensus, network structure, transaction, signature, hashing, and data placement in a blockchain, and the choice of each component affects the design and performance of that network. Consensus algorithm (*Garay & Kiayias, 2020*) is used to choose the person creating a block when it is added to a blockchain. The most popular consensus is proof-of-work used in Bitcoin and Ethereum. Delegated proof-of-stake (DPoS) has emerged as a favored consensus due to its scalability in *EOS.IO, S (2018)* and *TomoChain (2018)*. The voting process of DPoS is typically based on Byzantine fault tolerance. On the other hand, practical Byzantine fault tolerance is used mainly in private or consortium blockchains such as Hyperledger Fabric (*Androulaki et al., 2018*) and Quorum (*JP Morgan Chase, 2016*). The directed acyclic graph-based consensus is often applied on blockchain networks whose structure is a tree topology, such as IOTA (*Popov, 2018*; *Zhao & Yu, 2019*) and Hashgraph (*Baird, 2016*). Regarding the structure, a blockchain also can be a chain of blocks (*Nakamoto, 2008*; *Wood, 2018*) or trees (*Hoxha, 2018*), where each transaction is considered as a block. Transaction is used to store exchanged data including payment (*Wood, 2018*), data log (*Azaria et al., 2016*), or smart contract (*Christidis & Devetsikiotis, 2016*). In each transaction, a signature is embedded for authentication. The current technology used to generate signatures consists of single and multi-signatures (*Aitzhan & Svetinovic, 2016*), attribute-based signatures (*Sun et al., 2018*), and quantum resistance (*Perlner & Cooper, 2009*). The most popular hashing technique is SHA256, but there are under-researched ones including Scrypt (*Sarah, 2018*), Cryptonight (*Hill & Bellekens, 2018*), and X11 (*Marshall, 2018*). Regarding data placement, we classify it as on-chain or off chain(*Poon & Dryja, 2016*; *Chen et al., 2019b*). If data is written directly to the main blockchain, it is called on-chain data. However, if data is pre-processed in a private network first and then written later to the main blockchain, it will be off-chain data. In the rise of inter-blockchain communication, if a blockchain can accept tokens from other blockchains, it is classified as a cross chain (*Wang, Cen & Li, 2017*).

We figured out the research trends related to the blockchain by analyzing relevant works. In Swan's book (*Swan, 2015*), the author listed many challenges and limitations for the classification to map the existing researches on the blockchain. Although several challenges presented in Swan's work are reused in our taxonomy, we have a different view on the

classification of blockchain research trends. Thus, there are four main directions addressed in our viewpoint, including deployment, openness, confidence, and solutions. Studies in the deployment branch focus on improving efficiency in deploying blockchain networks. Typical researches for the group are (*McGinn et al., 2016*) with Bitcoin transaction visualization, and (*Rauchs & Hileman, 2010*) with benchmarking study. Meanwhile, energy is a hot topic that has attracted many scientists nowadays. Concretely, (*Wang & Liu, 2015*) suggested an economic model for getting high economic returns in consideration of the use of mining hardware with high computation-over-power efficiency and electricity price. The authors of (*Paul, Sarkar & Mukherjee, 2014*) calculated and showed how a new scheme could lead to an energy-efficient Bitcoin. Blockchain system deployment still causes difficulties for users, who must follow complicated guide documentation published on the Internet, especially open-source blockchain solutions. There are no tools or studies that try to provide easy deployment ways for this technology. Besides, for the scalability problem, there are also no works dealing with transaction storage, latency, throughput, size, and bandwidth of blockchain components in the manner of optimizing them in huge scale or global networks.

In research trends class, we also address the openness problem in current blockchain systems. (*Swan, 2015*) figured out that Bitcoin's APIs are quite challenging to use. On the other hand, the issues of transaction validity check and blockchain network analysis still have not been supported via APIs provided vendors (*Spagnuolo, Maggi & Zanero, 2014*; *Vandervort, 2014*). Although multiple chains, fork, and cross-chain are mentioned as the future of blockchain technology (*Lin & Qiang, 2019*), unfortunately, until now, no studies have brought forward potential solutions for them.

Confidence and security are emergent challenges since the arrival of blockchain technologies (*Mohsin et al., 2019*). *Bos et al. (2014)* stated that the use of elliptic curve cryptography (ECC), which is used to derive Bitcoin addresses to users, is insufficient and does not have the required randomness. *Mann & Loebenberger (2017)* suggested two-factor authentication for a Bitcoin wallet. *Lim et al. (2014)* analyzed the trend of security breaches in Bitcoin and their countermeasures. The authors also introduced some security countermeasures for individual users and safe Bitcoin transactions (e.g., a hardware wallet and a hardware authentication device). Although the strong point of the blockchain technique is data integrity, related studies are still concerned with improving this blockchain feature in order to prevent unauthorized data changes (*Bamert et al., 2014*). *Armknecht et al. (2015)* explained how to support security and privacy in the Ripple system, which is one of the consensus-based distributed payment protocols. For 51% attack problem, *Eyal & Sirer (2018)* introduced a selfish mine attack where colluding miners obtain a revenue more extensive than a fair share by keeping their discovered blocks private. The authors proposed a modification protocol which commands less than 1/4 of the total computation power. *Decker & Wattenhofer (2013)*, the propagation delay in the Bitcoin network is the primary cause for blockchain forks and inconsistencies among replicas, which was done by analyzing the blockchain synchronization mechanism. DDoS and scam are urgent problems of blockchain networks. *Vasek & Moore (2015)* investigated four types of Bitcoin scams (ponzi scams, mining scams, scam wallet, and fraudulent exchanges) by tracking

online forums and voluntary vigilantes. *Lim et al. (2014)* also addressed the DDoS attacks in Bitcoin.

Also, many other works focus on building novel blockchain solutions. Developing new blockchain platforms and cryptocurrencies are very typical examples of this research trend. The number of cryptocurrencies available over the Internet as of December 2019 is over 3,000 and growing (*CoinLore, 2019*). There are also many blockchain platforms developed over the last years (*G2 Crowd, 2019*). However, as addressed in openness trend, the interoperability (i.e., fork, cross-chain) of those platforms is still an open research topic, which has been concerned at present. In terms of solutions in this research trend category, bringing currency exchange mechanisms forwards in a secure manner is another urgent requirement of the blockchain user community. Meanwhile, creating new consensus models as well as mechanisms for the smart contract to enable blockchain technology application to resolve practical problems is a research trend that must be mentioned even in related work in the early stage.

In recent years, a significant number of blockchain applications have been developed by researchers and organizations (*Zheng et al., 2018*; *Nizamuddin et al., 2019*; *Xu, Weber & Staples, 2019*). Available applications belong to four main groups: (1) Transportation: is a sector where blockchain technology is creating a growing influence because it can create more efficient business operations in transportation companies, which are faced with tremendous demand from users to save costs and reduce delivery time. Applications in this groups relate to shipment tracking (e.g., *ShipChain, 2019*), autonomous vehicles management and usage (e.g., *DAV, 2018*), ride-sharing (e.g., *LaZooz, 2019*), and car renting (e.g., *HireGo, 2019*), and so forth. (2) Identification and authentication management is one of the great advantages that blockchain brings to authorities and users. Based on two fundamental features of the blockchain (i.e., distributed ledger and consensus mechanism), blockchain applications ensure reliable data and fraud prevention. In this group, there is a variety of applications, including citizen registration & identification (e.g., *Bitnation, 2019*), customer validation (e.g., patent of Barclay Bank *Marie, 2018*), degree verification & storage (e.g., educational platforms (*Li & Han, 2019*; *Chen et al., 2019a*)), and supply chain management (*Azzi, Chamoun & Sokhn, 2019*; *Gonczol et al., 2020*), and so on. (3) Public Service: is a domain that has received significant interest from the governments and organizations. An increasing number of governments in the world including the European Union, Estonia, Dubai are investigating advantages of blockchain when applied in their existing service systems such as land registry system (e.g., the project between the Sweden government and *Chromaway, 2019*), energy trading management (*Di Silvestre et al., 2020*), health care (*Chen et al., 2019a*; *Tanwar, Parekh & Evans, 2020*), and voting (*Khan, Arshad & Khan, 2020*). (4) Finance: is the domain that accounts for a majority of blockchain applications. It is not only the generation of the first blockchain application (i.e., bitcoin *Narayanan et al., 2016*) but also the origin of many key concepts in blockchain such as transaction, ledger, consensus, and signature. Therefore, it is understandable when many blockchain applications focus on this domain in order to address challenges in creating highly secured crypto-currencies, providing quick and direct payment services between individuals and organizations (e.g., Corda platform of *R3, 2016*), reducing insurance

fraud through sharing medical records (e.g., the project *Asia Insurance Review, 2018* of 15 insurers in India and two firms IBM and Cateina Technologies), and predicting markets candidly based on the participant of multiple users instead of a trusted individual (e.g., platform *Augur, 2019*), and so on.

## Certificate authentication using blockchain

Developing certificate authentication solutions using blockchain is categorized into identification and authentication management application presented in the taxonomy above. However, there were not many related works for the domain until now. *Grech & Camilleri (2017)* published a report of blockchain in education, which shows the potentials of blockchain to verify certificates. The authors pointed out that this technology can address existing limitations of paper certificates (such as the risk of forgery, loss, manual validity, certificate recall). Also, non-blockchain digital certificates (e.g., risk of forgery when a digital signature is not used, control of third-party certificate providers, risk of destroying electronic records). *Turkanović et al. (2018)* proposed a global higher education credit platform, named EduCTX. This platform is based on the concept of the European Credit Transfer and Accumulation System (ECTS). The prototype was developed using the open-source Ark Blockchain Platform (*ARK, 2019b*) and published in the form of open-source under MIT license on Github repository (*ARK, 2019a*). In (*Gräther et al., 2018*), the authors presented a Blockchain Education platform for issuing, validating, and sharing certificates. The platform aims to support counterfeit protection as well as secure access and management of certificates according to the needs of learners, companies, education institutions, and certification authorities. Blockcert (*Philipp, 2015*) is a system which can create issue, view, and verify credentials based on Bitcoin blockchain. The process of issuing a certificate is relatively simple, which requires the recipient's blockchain address, issuer's public key, and information on this certificate, such as the issue date. A small number of the university including MIT, the University of Nicosia, and the University of Birmingham have developed their systems based on Blockcert. In order to address some limitations of Blockcert, such as the increase of fee transaction, scalability issues of Bitcoin network, and retrieval ability of issued certificates, some certificate verification platforms were introduced (*Nguyen et al., 2018*; *Axel & Frankie, 2018*; *Abdul, Mahad & Waqas, 2018*; *Jirgensons & Kapenieks, 2018*). These systems make use of different blockchain platforms (i.e., Ethereum or Tangle). *Jeong & Choi (2019)* combined Blockcert with Bitcoin and Ethereum networks to build a recruitment management platform. This platform aims to make the recruitment and application process more comfortable and more flexible, together with increased trustworthiness.

Although existing studies allow educational institutions to create certificates and store the information on the blockchain network, these approaches do not provide mechanisms to record the training process of learners. Therefore, they cannot meet complex requirements in Vietnam, as introduced above. In our previous work (*Dao, Nguyen & Do, 2019*), we presented challenges and strategies when developing decentralized applications based on blockchain technology. By addressing the problems of data models, deployment scenarios, business processes, and consensus models using an example application, that work is a

guideline for enterprises or individuals, who want to apply blockchain to their information system. Although we also described the certificate authentication problem in the paper, we just stopped at introducing the application concepts without a detail discussion dealing with the educational certification authentication problem as compared with this works presented in this paper.

The number of studies on blockchain for the education field is still minimal. Few of them tend to be used in broad scope across different countries, but resolving a concrete education problem on a smaller scale in one country like Vietnam, the proposals could not be applied because too many changes must be made to meet the situations of that country.

## VIETNAMESE EDUCATIONAL SYSTEM AND CERTIFICATE AUTHENTICATION PROBLEM

In Vietnam, the Ministry of Education and Training (MOET) takes responsibility for all education and training activities at the national level. The MOET is divided into many separate departments, which are the most important are those responsible for almost all educational types. Structurally, the primary education system in Vietnam has complete all levels, including pre-school, primary, secondary, high school, and higher educations. Besides full-time education, there are in-service (part-time), and distance learning types in Vietnam. However, because achieved education quality does not meet the expectation of employers, part-time and distance learning types have been reduced on the training size. Although MOET plays a central role in education management, many training institutions fall under other ministries and government agencies. For example, vocational and technical education is under the control of the Ministry of Labor, War Invalids and Social Affairs (MOLISA), and art schools and conservatories are under the management of the Ministry of Culture, Sports and Tourism (MOCST). Moreover, there are many local universities, schools, or other types of education organizations, which are controlled and funded by provincial and district governments. In several aspects, the multiplicity of actors can result in duplication and confusion in management.

In terms of educational institution classification, there are three different types as follows. *Public* - the institution is managed by a public authority such as MOET, provincial, district, or commune level. The state provides all operating costs of the public institutions. *Semi-public*—the facility is owned by the state and managed by a public authority, but student fees cover all operating costs. *People-founded (private)*: people-founded institutions are owned and managed by non-government organizations. In Vietnam, there are also many training centers, which offer a vast number of short courses like English, informatics, and other specialties. Those centers often are managed by private associations such as trade unions, cooperatives, youth organizations, and women's associations. Some others are under control by companies or even individuals. With a large number of these centers, the management, operation audit, as well as training quality evaluation of government agencies, is quite tricky. As a result, certificates issued by the centers usually do not have quality assurance and origins.

Technically, no one can ensure the trustworthiness of diploma or certificates issued by the educational institutions except themselves. Although several of them update the lists of graduates on their official websites, most institutions do not publish the certificate and diploma data of learners. On the other hand, to authenticate precisely a learner's certificate, not only the certificate database but also the training process data of the learner also need to be provided by educational institutions. The training process data is a critical information source, which confirms learners fully attended and deserve their issued certificates. In Vietnam today, there are no common databases managed by MOET or other national agencies, which provide all data related to learners, including both certificates and training process results. This fact leads to the problem of fake degrees, diplomas, certificates that is quite popular in Vietnam. There have been several scandals about students, or even officials obtaining degrees, certificates from dubious universities and educational centers. A significant number of people are being arrested for printing fake degrees, but the practice thrives, nevertheless (*Vietnamnet Journal, 2018*).

Until now, there is no efficient way to authenticate certifications in Vietnam. Thus, when someone would like to look up the education profile of a person, he/she cannot find the certificate database, or even if it exists, the certificate data provided by the institution may not be reliable because there is no training process history of the objects. Hence, the need for a system that can authenticate these documents and training processes is urgently required in Vietnam nowadays. In summary, the situation of training and education in Vietnam has the following distinctive characteristics as compared with other countries. These are also prerequisites for building a certificate authentication solution using blockchain technology for Vietnam.

1. The national education structure is asynchronous because not only MOET manages and controls training institutions but also other government, as well as social organizations. Thus, there is no consensus in certificate forms, as well as a mechanism in controlling the training process and quality. Hence, a certificate authentication system developed for Vietnam must define *unified business processes* that allow reading and writing data on blockchain ledgers simply and effectively.

2. Almost all decentralized applications today have exploited permissionless blockchain (e.g., Bitcoin or Ethereum) to store transaction data. With these networks, these applications will depend on complex consensus algorithms (i.e., proof-of-work, proof-of-stake) that may be unsuitable with the application features, especially in the case that digital token does not make sense. The certificate authentication system is one of the application kinds, which does not maintain any crypto-currency. Moreover, in general, in Vietnam, the system must be controlled by MOET with the role of public management to ensure training quality. Hence, *deployment model and consensus technique* are also critical factors for developing the certificate authentication solution for Vietnam.

3. Because the number of training organizations in Vietnam is quite large and belongs to different educational types as well as training scale, the proposed authentication system should have *an open architecture* that enables different organizations and actors (i.e., universities, schools, managers, teachers/lecturers, students) to join in the future.

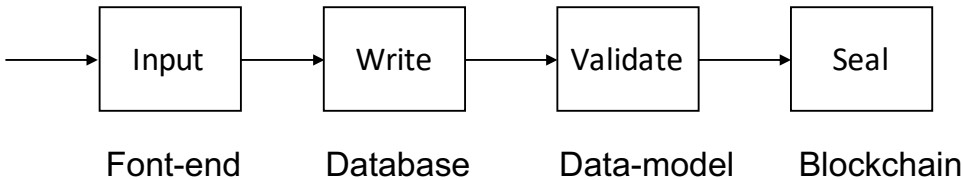

**Figure 2  VECefblock four-phase design.**

4. Although several existing IT solutions like credit management system, and school record management systems have been deployed in universities and schools in large cities in Vietnam, these systems do not comply with any data structure standard. Hence, the proposed system must also define *single unified data structure*, which helps data sent from existing applications to the certificate authentication system in an easy manner.

5. Besides, with a larger number of users, data privacy, blockchain network performance, data query mechanism are also important factors that should be considered in the system design process.

# DESIGNING SYSTEM

In this section, we present our designs to build the VECefblock system with a focus on its components and operation processes.

## VECefblock overview

For VECefblock, we propose the input-write-validate-seal architecture that improves and enhances the trust and openness of traditional educational management systems in schools and universities. The proposed architecture is called the four-phase style. Meanwhile, the existing traditional architecture has only two phases, including input-write. Figure 2 illustrates the difference between the two approaches.

- *Input:* It is the same in both architectures, where educational data, such as assessments, student identification, and certificate information, are input from a desktop program or web-based portal. Note that educational institutions control those input applications. The possible actors of such applications are education departments, school/university managers, and teachers.

- *Write:* In the traditional approach, input data is written to a so-called local database running on the same desktop or web server with the applications. In our approach, however, data is written to both the local database and blockchain. The input data is converted through our data mapping structure.

- *Validate:* After the input data is matched with the data structure of VECefblock, its format and signature are checked. That input data becomes a transaction in VECefblock. Note that the electronic signatures are provided and ensured by national centralized authentication service providers (i.e., CA providers). By identifying the system's actors, data stored on the blockchain is trustworthy.

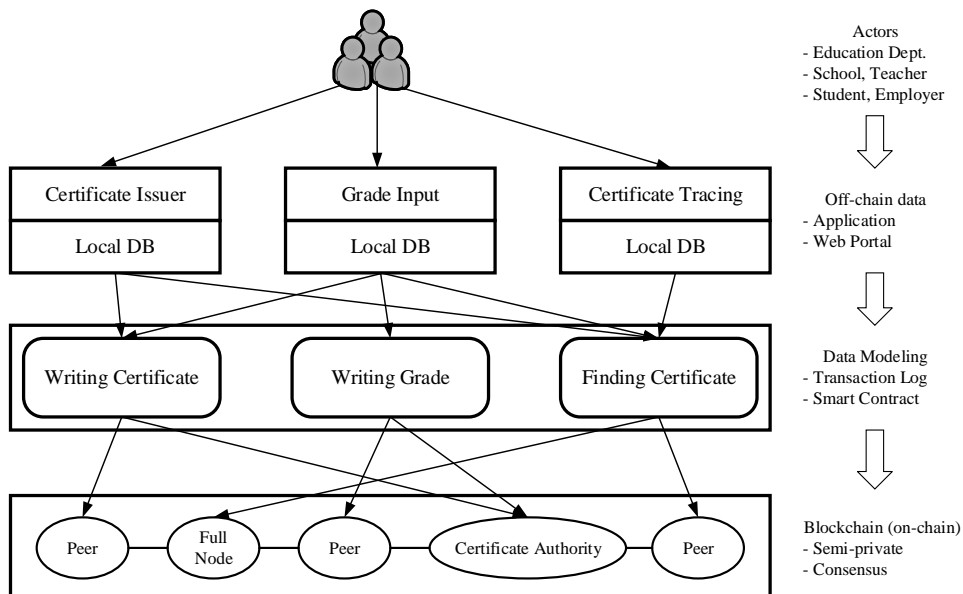

**Figure 3** **System architecture.**

- *Seal:* Validated transactions are put into a block, and a new block is created to seal those transactions. After sealing, it is almost impossible to change the data, and any modification can also be recorded on the blockchain.

VECefblock consists of three main layers that contain our proposed four-phase architecture. Figure 3 shows actors on the top and three layers having functionalities of off-chain writing (local database) (*Poon & Dryja, 2016*), data modeling, and on-chain writing (blockchain), respectively.

- *Application layer:* The *input* and *local write* phases belong to this layer that covers all interactive applications running at education institutions. We call its functionality off-chain data since those input data is not reflected in VECefblock.

- *Data modeling layer:* Before being written to the VECefblock blockchain, data from the application layer must be converted to our data structure. This layer contains a data mapping structure, such as writing certificates, writing grade, and finding a certificate. The detail of the data mapping model is described later in the next section.

- *Blockchain layer: Validation* and *sealing* happen on the blockchain. The typical components of a blockchain include peers, full nodes, and certificate authority. VECefblock could be deployed on different blockchain platforms, such as Hyperledger Fabric (*Androulaki et al., 2018*), Quorum (*JP Morgan Chase, 2016*), and EOSIO (*EOS.IO, S, 2018*). However, because the certificate authentication is a state management procedure, it requires high-level security (according to Vietnamese network security law, all Internet service providers must store data on servers located inside Vietnam). Besides, state agencies always retain the right to issue educational certificates. Therefore, in our design, we recommend deploying a permissioned blockchain network using

open source platforms like Hyperledger Fabric. Normally, for each deployment side of Fabric (e.g., university, education organization), three peers, and one system certificate authentication (CA) node are minimal deployment patterns.

## Deployment and consensus model of VECefblock

In order to use VECefblock in practice, considering deployment, including the type of blockchain network, the performance of transactions, and the metric of used RAM and CPU, is essential. Based on the real situation in Vietnam, a permissioned blockchain network is more relevant since issuing certificates needs to be checked by MOET. Because it is not necessary to issue certificates in real-time, the network and performance of transactions could be at the intermediate level.

Since a permissioned blockchain network is used to deploy VECefblock, a consensus model could be simple, such as Practical Byzantine Fault Tolerance. Participants must register in advance to join the private network. MOET has an administration role to assign who can read data from and write data to the blockchain. Educational institutions must validate transactions before the Ministry's decision. Note that the Ministry checks whether the signature of an educational institution is authenticated, then whether it can issue the certificate. For example, some universities can issue a bachelor's degree, but some cannot confer a master's degree.

## Blockchain-based system business processes

Figure 4 describes the operation processes of our system. There are several actors, who participate in VECefBlock, including education institutions (e.g., schools, colleges, universities), learners (e.g., pupils, students), teachers (e.g., professors, lecturers, teaching assistants), and certificate checker (e.g., enterprises, employers).

_Education institutions_ play the role of defining training smart contract conditions, which point out the number of credit courses that must be collected by a student before he/she can receive a certificate. In other words, this contract defines education programs with a set of courses. _Learners_ actor must complete the contract conditions through collecting credits of courses. When a learner enrolls in an education institution, a local ID will be issued for the learner (e.g., pupil or student ID) by the institution. As system design, each institution has a local database that provides a data repository for its educational management system. This design suits the most situation in Vietnam, where each university and school has its education management system. The local learner ID thus is a unique identification in the system of that institution. However, to ensure the ID of the learner is unique in VECefblock, we select national IDs provided by the government instead of using local IDs. In case that an educational institution does not store national IDs of leaners, we will make use of learner ID, institution name, timestamp (at the time of registering ID data). These values will be hashed together to create the identification key of the learner in the blockchain network.

In our operation process, the learner will do examinations assigned by _teachers_ actor. Once the learner passes exams, the teacher will write course results into a local database of the education institution. Through the data model presented hereafter, the educational data will automatically be transferred to our VECefBlock.

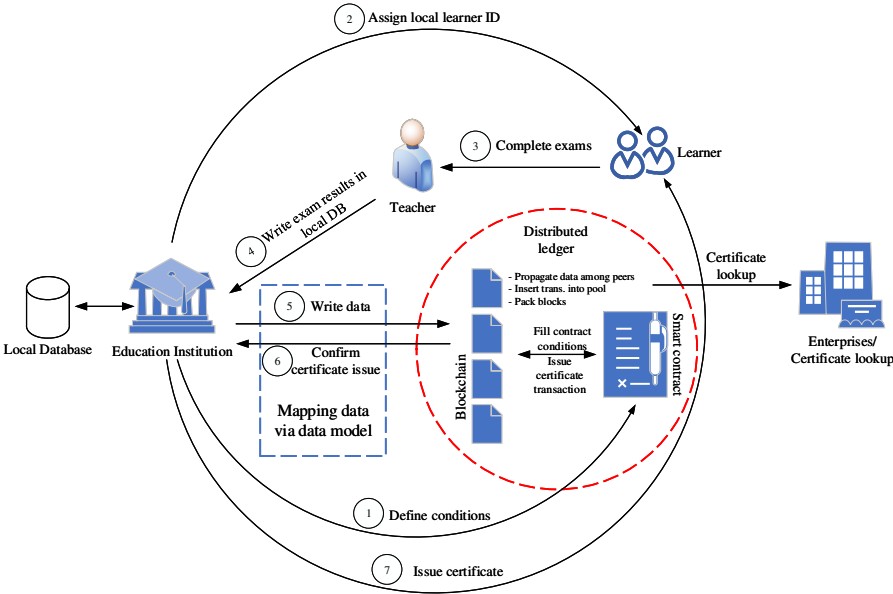

**Figure 4** **VECefblock business processes.**

Once all the smart contract conditions related to credit records are met, a transaction of certificate issue is automatically created and stored on the blockchain network. With our data mapping structure, the certificate issue information will be sent to the educational institution with a key of issued diploma. Based on the advantages of blockchain technology, the certificate information is propagated and verified by blocks stored in many different peer nodes. Therefore, the information can be trusted and cannot be changed by third parties. Based on the certificate information on the blockchain network, the institution will issue both certificates in the form of document and electronic (certificate codes) for learners.

For enterprises who would like to look up a certificate, they can query data stored in blockchain transactions through certificate codes issued for learners. On the other hand, to secure the writing course results processes, we use encrypted private keys in the following manner. During writing the exam result data, teachers must embed their private keys. The educational institution also uses its key (i.e., electronic signatures provided by national CA organizations) combining with the teacher keys to write data in the blockchain system. This mechanism helps prevent fake transactions during writing data of teachers on the local institution database and from the local database to the blockchain network.

## Data structures

Figure 5 introduces data structures used for the data transformation process in the system. This process consists of three elements (i.e., input data in the form of either relational

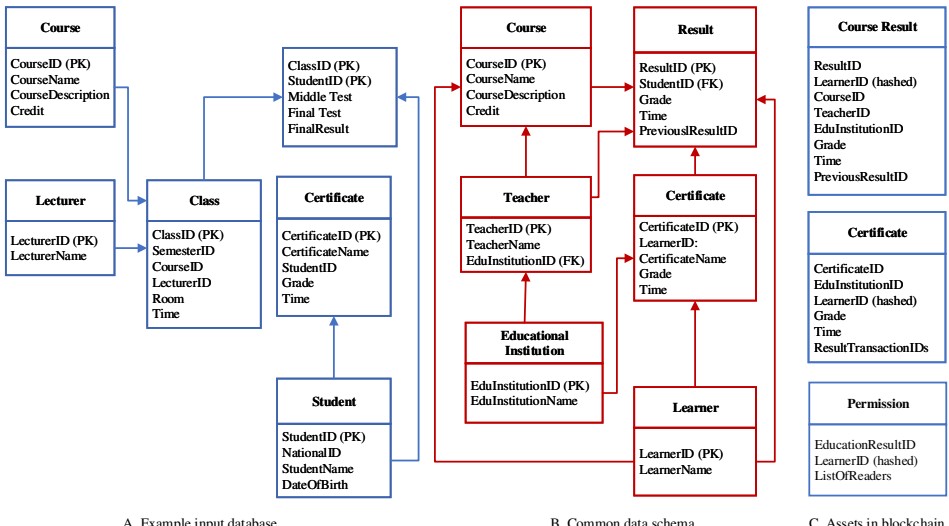

A. Example input database      B. Common data schema      C. Assets in blockchain

**Figure 5**   **Data structures.**

database or flat files such as CSV, Excel), a mediated data schema, and three different types of assets for data storage and permission definition.

- *Input data:* The input of the data transformation is a relational database or flat files that are generated from a database. Educational institutions may have heterogeneous data schemata, which are different from their structures and naming. However, they should store information on learners, teachers, and learners' results.

- *A mediated data schema:* is used for consolidating different local data schemata. Based on this mediated schema, the input data will be transformed into a consolidated representation of structure and name. This schema contains only necessary information used for validating learners' certificates. Therefore, we identify six important tables in this schema, including educational institution, teacher, learner, course, result, and certificate. Depending on what requirements may be arisen in the future, we can extend more tables. To conduct the data transformation, we are currently using SQL server integration services (*Microsoft, 2017*).

- *Assets on blockchain network:* In Hyperledger Fabric, an asset is a collection of key-value pairs, which is used to represent anything of value such as real estate, hardware, and contract. In VECefblock, we define three different assets: (1) *course result* contains grade of each course, (2) *certificate* stores information of learners' certificate, and (3) *permission* defines control levels of different users to educational results.

## Data privacy

Centralized systems typically store a huge number of information on users, but they have little control of the stored data and how the data is used and passed to a different organization. In our system, we allow learners to own their data and can grant permission to other users to access the data. To this end, we make use of *Permission* asset, which

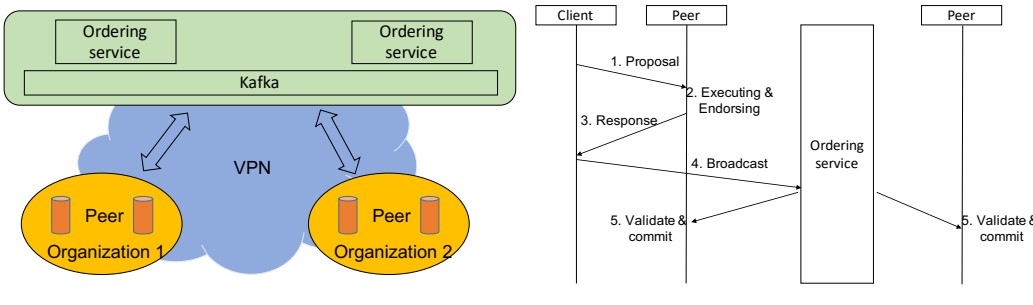

**Figure 6** **Hyperledger Fabric deployment.**

defines access levels of different users for educational results stored on the blockchain network. When a transaction of educational result (regarding to *course result* or *certificate* asset) is created, our system will generate a *permission* asset. A *permission* asset stores three important information, covering (i) ID of education result asset, (ii) ID of Learner, which determines the owner of the corresponding educational data and, (iii) a list of users who can read the data. The learner can manage this list by removing a user or adding a new user. However, the education institution - the default user in this list cannot be removed by the learner.

## EXPERIMENTAL METHODOLOGY

We use Hyperledger Fabric (*Androulaki et al., 2018*) to implement our VECefblock blockchain, and its application is written in Javascript using NodeJS. In order to evaluate VECefblock implementation, we measure throughput and latency as the primary performance metrics. Throughput is calculated by measuring how many transactions are committed to the ledger of VECefblock blockchain each second. Latency is taken time from sending a transaction to being committed. Note that although there are several kinds of latency in Hyperledger Fabric, such as endorsing, broadcast, ordering, and committing time, we measure the total latency. We use a dummy workload for each experiment.

### Hyperledger fabric deployment
There are three main components: organizations, ordering services, and Kafka (*Kreps, Narkhede & Rao, 2011*) in the deployment of Hyperledger Fabric on a cluster. While educational institutions manage peers, the ordering services are managed by the Ministry. We illustrate its logical architecture in Fig. 6A in which ordering services and Kafka are grouped on the top. There are two organizations on the Figure, and there are several peers on each organization. All components are connected by using IP addresses and ports. Its network can be either LAN or VPN.

An organization includes peers having the same policy and authentication. The number of peers in an organization is not limited. The peer thus plays the role similarly to either a miner or a light node in the Bitcoin network. An organization can be seen as a supernode.

A peer has the right to authenticate transactions, and the distributed ledger is maintained in each peer. In order to deploy our application on Hyperledger Fabric, developers need to install a chain code (a.k.a smart contract) for peers. Based on the chain code, a peer can verify whether a transaction is valid.

An ordering service plays a role in creating a new block to the distributed ledger. Typically, there are several ordering services in order to ensure fault tolerance when creating blocks. Hyperledger Fabric on IBM recommends five ordering services, but in this deployment, we installed two ordering services. Note that all ordering services create a new block at the same time. Kafka is used when deploying Hyperledger Fabric, which can be seen as a consensus mechanism because it requires nodes to agree on the same order of transactions. Ordering services first sends transactions to Kafka and receives them from Kafka in the same order.

### Transaction flow

Figure 6B shows how a transaction is processed. There are three actors in the process: clients, peers, and ordering services. A client plays a role in sending transactions to a specific peer in an organization. After that peer confirms that the received transaction is validated and executed, it is endorsed by that peer and sent back to the client. Next, the client will broadcast the endorsed transaction to ordering service. After a configured time, a block is created by ordering services, and that endorsed transaction is added to that block. The newly created block is sent to all peers, and each peer must update its ledger.

## EXPERIMENTAL RESULTS

The main goal of our experiments presented in this section is to evaluate the operating performance of VECefblock to show the operability and feasibility of the proposed system under the real conditions. The tests thus are an important reference for Vietnamese state authorities in applying blockchain technology to the fake certificate as well as other social problems in Vietnam today. In this direction of evaluations and tests, we run our experiments on a Hyperledger Fabric cluster having two organizations, two ordering services, and a group of three Kafka Docker nodes. There are two peers and one ordering service on each organization. We use three m5.2xlarge instances (*Amazon, 2019*) on Amazon EC2 to host our cluster. Each instance is equipped with eight vCPUs and 16 GB of memory. On our cluster, Kubernetes (*Brewer, 2015*) is used to manage Docker images. We employ Hyperledger version 1.2. We conducted three types of experiments: measuring pure TCP performance, writing grade transaction to the distributed ledger and reading grade and degree from the ledger, and impact of resource allocation on performance. We measure throughput and latency for each experiment. Each parameter configuration is run three times, and their average values are used to draw figures.

### Pure TCP performance

The purpose of this experiment is to show the network performance of the cluster is fast, and the latency of node communication is small enough to be removed from the total latency measured in the Hyperledger Fabric-based blockchain network?s experiments. We

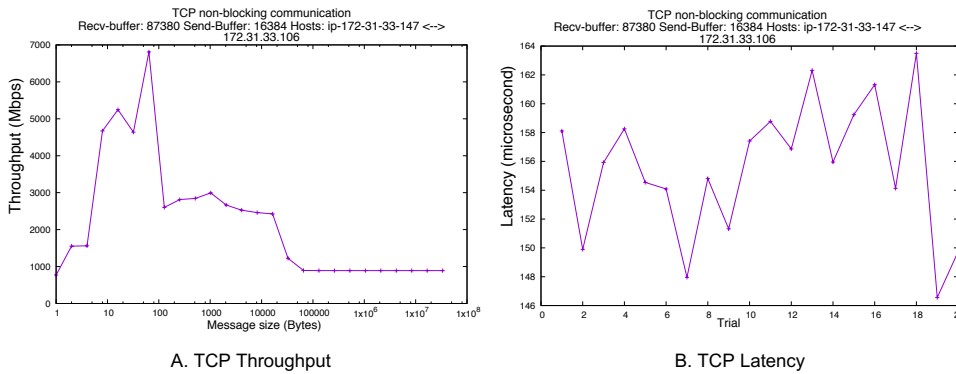

**Figure 7** **Pure TCP performance.** (A) TCP throughput. (B) TCP latency.

configured our testbed on Amazon EC2 with two new Ubuntu instances. Note that there is no other traffic except our experiment.

The network is considered fast among nodes on Amazon EC2, and its peak can be up to 10,000 Mbps. In order to measure throughput, we increase message size in each configuration and get the throughput. For the latency, we fix the message size at 3 KB since the transaction size of Hyperledger Fabric is approximately three KB also. We base on HPCBench (*Huang, Bauer & Katchabaw, 2005*) to do the benchmark, and non-blocking communication is used to measure the throughput and latency. For the TCP parameters, receiving and sending buffers are 87,380 and 16,384 bytes, respectively. The TCP version on Ubuntu is Cubic. Figure 7A shows that the peak throughput is 7,000 Mbps when the message size is 64 bytes. The throughput decreases sharply to 1,000 Mbps when the message size is bigger than MTU size (1,500 bytes). At the message size of 3 KB, the throughput is 2,500 Mbps. The latency of 3 KB message size is shown by Fig. 7B, and the average time is 150 microseconds. This latency is relatively small if we convert to the second unit.

## Writing and reading performance

The purpose of this experiment is to check how fast our VECefblock blockchain is in the aspect of writing student grades to the ledger. Each grade given by a teacher is considered as a transaction. More transactions are processed, better performance is obtained. The transaction rate is defined in terms of how many transactions are sent to the VECefblock blockchain each second. We change that transaction rate in each configuration.

Figure 8A shows that the peak throughput in our VECefblock cluster is 8,000 transactions per second (tps) when the incoming transaction is eight. When the transaction rate is more than 32, the throughput is almost unchanged, and the limitation of the cluster is revealed. At a small transaction rate, the throughput is high since all incoming transactions can be packed into a block. This figure also shows that the latency has been gradually increasing when the transaction rate is less than 16. The reason is that there are more transactions waiting for packing into a block, and we used only a few clients sending transactions to peers. The bottleneck of broadcast and response might happen at the clients.

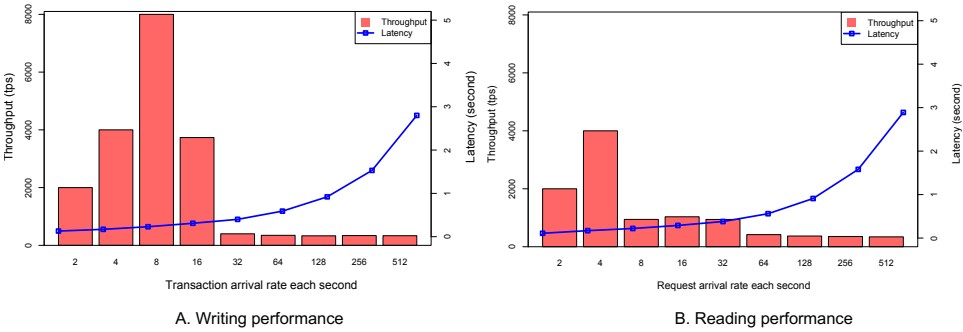

**Figure 8 Reading and Writing performances.** (A) Writing performance. (B) Reading performance.

This experiment aims to verify the performance of transaction retrieval from the VECefblock blockchain. We find certificate information based on its unique ID, and the algorithm is the brute force by checking all blocks in VECefblock blockchain. Figure 8B shows that the peak throughput of searching is 4,000 tps, and the latency is increased when the transaction rate is more than 32. The throughput is unchanged at 350 tps when the transaction rate is more than 128. The reason is that a brute force of algorithm is used. Hence, there is a limitation of searching at each peer. Note that the searching request can be sent to any peer on the VECefblock blockchain. Our proposed blockchain network is faster than several Hyperledger Fabric performance in (*Kuzlu et al., 2019*; *Basumatary & Peri, 2019*).

## Impact of resource allocation

The purpose of this experiment is to show whether the scalability can be obtained or not when more resources are allocated to each peer node. We fix the transaction rate and change the number of virtual CPUs and the amount of RAM for each configuration. The range of virtual CPUs is from one to eight cores, and the size of RAM can be changed from one GB to 16 GB. Figure 9 shows that when the number of vCPUs is increased from two to four, the throughput goes up, and the latency is decreased. The observation happens similarly at three configurations of the transaction rate of 128, 256, and 512. However, the size of RAM does not affect too much to the performance.

## CONCLUSION AND FUTURE WORK

This study was carried out in the emergence of blockchain technology with cryptocurrencies like Bitcoin and Ethereum. However, the decentralization and data immutability of the technology brings many advantages that could change business operations of existing systems today. In this paper, we described the certificate authentication problem in Vietnam, then we proposed our prototype called VECefblock, which is a blockchain-based application dealing with the problem of the fake diploma mentioned above. Although the proposed VECefblock was developed to suit Vietnamese conditions, the principle designs also can be inherited in certification authentication systems in other countries, especially VECefblock businesses. The contributions of this study can be listed as follows.

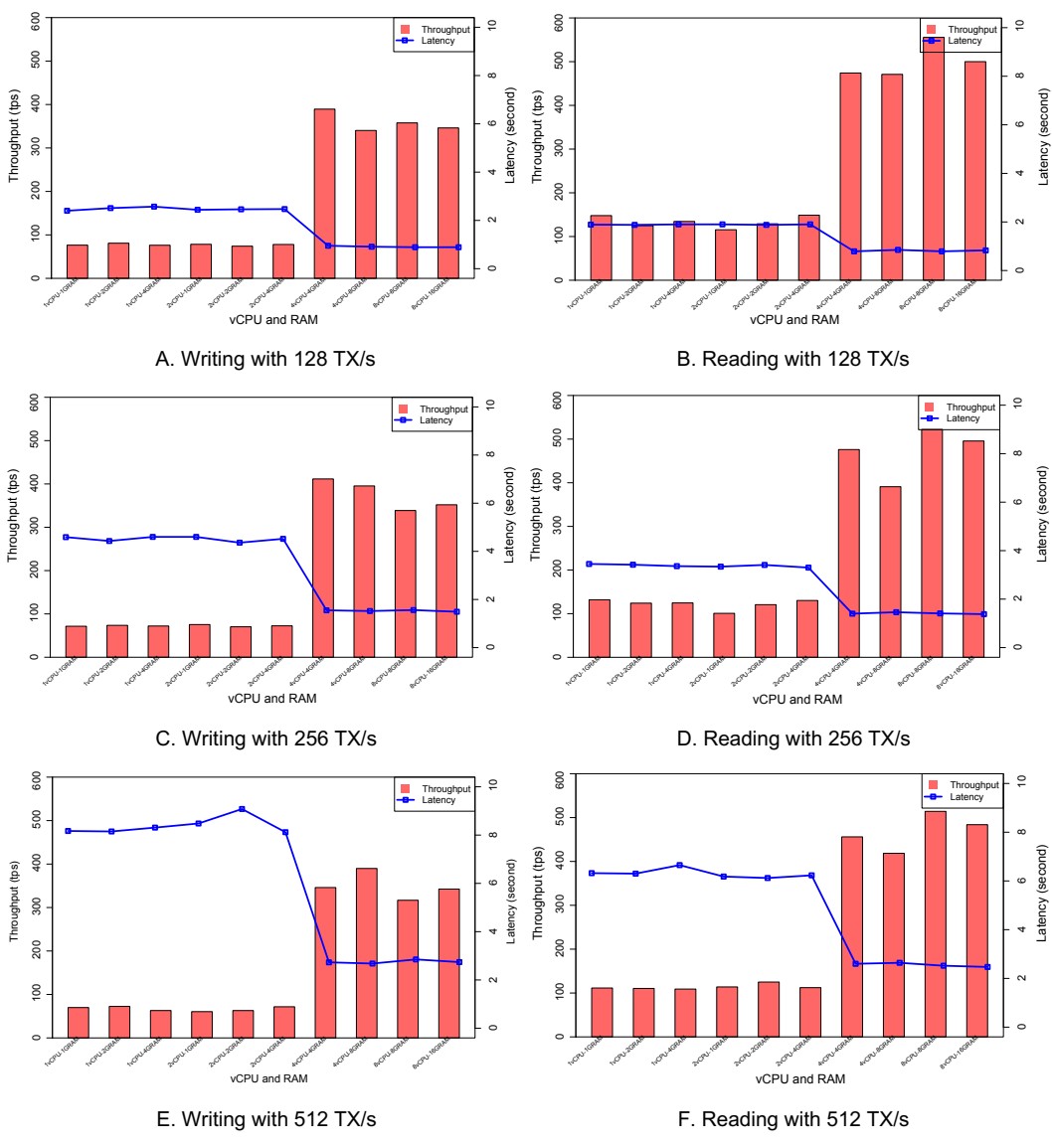

**Figure 9** **Impact of resource allocation.** (A) Writing with 128 TX/s. (B) Reading with 128 TX/s. (C) with 256 TX/s. (D) Reading with 256 TX/s. (E) Writing with 512 TX/s. (F) Reading with 512 TX/s.

1. Summarizing state of the art of blockchain technology in different aspects, which is shown by a detailed taxonomy. Analyzing the existing blockchain-based applications applying to education in general as well as certificate authentication.
2. Discussing the emerged fake diploma problem in Vietnam and pointing out the potential approach using distributed ledger technology.
3. Identifying steps to develop a blockchain-based application. Thus, developers can follow these steps to build their decentralized blockchain applications.

4. Developing a VECefblock prototype - a blockchain-based application with architecture, business processes, transaction flows, and data mapping structure. The prototype suits Vietnam's conditions in training and certificate issue.

5. Deploying and testing VECefblock in a cloud environment using Hyperledger Fabric with several performance stress experiments.

6. The work also proves and promotes the application of blockchain technology in Vietnam.

There are several future works based on this initial research. We plan to scale up the VECefblock system with more peer nodes that are deployed at different locations (e.g., schools, education management agencies at districts, and cities). Also, we will focus on building an effective query data mechanism for the proposed systems to enable semantic search on blockchain-based data transactions. We are also interested in the operability of VECefblock services when deploying it on different sides by carrying out performance analysis tests in the future.

### Funding
This research is supported by the Vingroup Innovation Foundation (VINIF), project code VINIF.2019.DA07. The funders had no role in study design, data collection and analysis, decision to publish, or preparation of the manuscript.

### Grant Disclosures
The following grant information was disclosed by the authors:
Vingroup Innovation Foundation (VINIF): VINIF.2019.DA07.

### Competing Interests
The authors declare there are no competing interests.

### Author Contributions
- Binh Minh Nguyen conceived and designed the experiments, prepared figures and/or tables, authored or reviewed drafts of the paper, and approved the final draft.
- Thanh-Chung Dao performed the experiments, performed the computation work, prepared figures and/or tables, authored or reviewed drafts of the paper, and approved the final draft.
- Ba-Lam Do analyzed the data, prepared figures and/or tables, authored or reviewed drafts of the paper, and approved the final draft.

### Data Availability
The code of the system is available at GitHub:
https://github.com/bk-blockchain/hyperledger-fabric/tree/master/src.

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
