# Peer review of "Towards a blockchain-based certificate authentication system in Vietnam"

_PeerJ Computer Science, doi:10.7717/peerj-cs.266_

## Round 0.1 · original submission · Major Revisions

Please revise the paper thoroughly according to the comments of the reviewers.

Reviewer 1 ·

Basic reporting

This study aims to solve the problem of forgery of certificates / graduations, a social problem in Vietnam, through integrated business services that authenticate and verify between government agencies, various educational institutions and participants through blockchain technology.

Experimental design

In the data structure, the common data schema that serves as the interface between the client's input data and the blockchain network's assets is designed to secure interoperability that can be linked with other blockchain networks in the future.

Validity of the findings

However, for the performance measurement results, the network performance measurement results according to the message size in the block, the number of transactions, the throughput statistics according to the number of virtual processors and the memory, and the latency statistics were presented and analyzed. It is also necessary to evaluate the performance according to the number of transactions, and there is no analysis in terms of services. In addition, although the consensus model is simple unlike the open blockchain due to the nature of the licensed blockchain, the proposed service does not know how to implement the consensus model by performing the performance evaluation without the specific consensus model. Finally, the proposed service is questionable whether it can be called a blockchain in that the state (Ministry of Education) has control over blockchain network configuration nodes or participants (recentralization) through a licensed blockchain.

Additional comments

Please make revisions as indicated

Reviewer 2 ·

Basic reporting

This paper proposes a blockchain-based certificate authentication system for managing all the educational diplomas in Vietnam. Additionally, this system is deployed using Hyperledger Fabric blockchain infrastructure and the paper presents experimental performance results based on this infrastructure. Apart from this core contribution of this paper, it also summarizes state of the art of blockchain technology in different aspects.

Overall, the paper is very well written and presents interesting and technical sound results. There are only some minor comments that could help the authors to improve the presentation of their paper:

1) In page 6 lines 234-247, the contribution of the paper could be moved in the Introduction section and somewhere at the end of the section (page 2, line 74-75).

2) It is good to mention, possibly in the Conclusion section, that the proposal system is generic and it can be applied to any country and not only in Vietnam.

3) The Figures 7, 8 and 9 should be resized because the text on them is very small for the readers.

4) In Tables 1 and 2, the numbers 2, 4, 8, …, 512 are huge (font size) compared to the rest of the text in them.

5) The authors, if they want, could also add a comment about the cost of Hyperledger Fabric per school or university (or even for MOET) and if it is still expensive to apply a blockchain infrastructure in a real setting.

Experimental design

no comment

Validity of the findings

no comment

·

Basic reporting

The paper presents an interesting application of the blockchain technology to provide notarization and transparency for skills certificates, and I enjoyed reading it.

The authors focus on the Vietnamese context and aim at providing a solution to the anticounterfeit of school and academic certificates. One of the most exciting aspects of the paper is the idea of using the blockchain not only to notarize and provide transparency related to the certificate but also include the learning process. Including the learning process permits to provide a logic for automatic issuing of the certificate once all the courses have been passed and certain conditions are met.

The background to show the context is clear, and it is mainly focused on the Vietnamese education system. It is not clear why the authors focused on the Vietnamese case. Although Vietnam has some conditions that make the system more appealing and useful, we consider that the tackled problem is general and can be considered world-wide, despite in some nations is less relevant.

The literature references are sufficient, and there are more than 10 references related to the use of blockchain for skill certification. The context is clear, and there are several references to the specific Vietnamese condition.

Experimental design

As for the results, the information provided by Fig. 7 appears to be completely independent of the application and from the blockchain as it is related to pure TCP performance. Furthermore, it is not clear what are the conditions on which the measures are taken, is there other traffic or it is only the traffic of the network_? What are the TCP parameters? Which TCP version?

Validity of the findings

The main contribution of the paper is related to the proof of concept of the application of blockchain for certifying skills.

Additional comments

The paper describes the steps for designing a blockchain-based system for providing transparency and trust in handling certifications of skills. However, the actual implementation details are not provided. The authors state that one of their contributions is the analysis of blockchain situations in the aspects of technologies, classifications, functionalities, research trends, and applications. Still, the paper has not the structure, nor the broad vision of a review paper, despite it clarifies several exciting elements, especially those reported in Figure 1.

Figure 1 is interesting and provides a proper classification of blockchain technology. Still, some elements are not consistent (e.g. a classification by chain openness contains a single chain, cross-chain and side chain, which is more related to the number of blockchains involved and their connections).
Furthermore, DAGs are reported as a kind of consensus, while they refer to the data structure, and we suggest removing ‘others’ from the figure, except there is a specific element to add.
Under the element ‘signature’, we would recommend considering single and multi-signatures.

Figure 2 reports the four-phase design of the proposed approach. It is not clear what is the trust model, especially regarding the validation phase. If the validation phase is just a formal validation of compliance to a specific data-model, it is ok and easy to perform. How are the skill certificate validated? The scholars can be untrusted, but what about the schools/universities/ private bodies? Should they be trusted as data flow in their database before being written in the blockchain? What it happens if a novel certificate 'appears' on the database? Is that included in the blockchain? What is the validation policy? Are the databases of the schools considered trusted?

Figure 4 reports an ‘issue certificate’ action. Is the certificate in the form of a document that can live outside the blockchain? In this case, where is the trust? I understood the certificate is issued on the blockchain, so what does this arrow represent?

Figure 6 shows only one ordering service. Please, consider showing more ordering services form multiple institutions and the message exchange between them.

Figures are relevant, high quality but are not always well-described. Furthermore, their relevance is reduced by the companion tables, which contain the data that are plotted in the figures. For example, table 1 and 2 provide the same information as Figure 8, tables 3 e 4 present the same results of Figure 9. We advise using tables or figures but not both.


The paper contains the sentence “Since a private blockchain network is used to deploy VECefblock". Private or permissioned? Furthermore, transactions are said to be validated by educational institutions before the Ministry decision. What is the Ministry decision? This part is not clear. If the Ministry can decide what can be written on the blockchain, then it is a central trusted authority, and a database should be used instead of a blockchain. Is the Ministry only able to decide which are the recognized educational institutions that can write on the blockchain?
It is not clear how the designed system prevents the ‘garbage in garbage out’ problem. What if the educational institution is not trusted and ‘sells’ a certification? Please, consider explicitly stating if the educational institutions (including the private ones) are trusted or not.
To ensure the uniqueness of the learnerID, the authors suggest hashing “the learner ID, institution name, timestamp (at the time of registering ID data) is hashed together as the identify key of learner in blockchain network”. So, there is a learnerID before the hashing? How is this id provided? Furthermore, adding the timestamp related to the student registration may result in a duplication of entries for one student. For example, a student is identified today in a school has an id H1 H(studentX | timestamp_today) and another id H2 if it is registered tomorrow H(student | timestamp_tomorrow). This method could provide a situation where one student is identified by multiple IDs, which is not what is generally desired.

The English used in the paper is good, but some sentences could be improved (e.g. the sentences highlighted in purple in the attached document). Some typos (e.g. Crytography) and grammatical mistakes are present, and some sentences are difficult to read. Furthermore, please, consider substituting the term 'blockchain engine', with the blockchain platform. In facts, this wording may be misleading, even if it applies to specific cases (e.g. where the blockchain is based on virtual machines), in general, it is not a valid wording.

As for the results, the information provided by Fig. 7 appears to be completely independent of the application and from the blockchain as it is related to pure TCP performance. Furthermore, it is not clear what are the conditions on which the measures are taken, is there other traffic or it is only the traffic of the network_? What are the TCP parameters? Which TCP version?

---

## Round 0.2 · Minor Revisions

Please make revisions based on reviewer#1's comments. I trust you will be doing this corrections so that an additional round of review can be omitted.

Reviewer 1 ·

Basic reporting

The authors should go through paper thoroughly and check for spelling and grammar issues for English as none of the authors are native English speakers.

Some key references are missing, please try to add some recent findings in this area from 2019 and/or 2020.

Experimental design

It is improved from the original version

Validity of the findings

Seem fine to me, better than original paper

Additional comments

Overall paper is improved and seems to be better than original. All my concerns were addressed in the revision as I was an original reviewer

Reviewer 2 ·

Basic reporting

The current version of the paper is ready for publication.

Experimental design

No comment

Validity of the findings

No comment

---

## Round 0.3 · accepted · Accept

The authors have fully addressed any remaining comments.